# Can satellite altimetry observe coastally trapped waves on sub-monthly timescales?

Marcello Passaro<sup>1</sup>

<sup>1</sup>Deutsches Geodätisches Forschungsinstitut der Technischen Universität München, Arcisstraße 21, 80333 Munich, Germany. **Correspondence:** Marcello Passaro (marcello.passaro@tum.de)

Abstract. Coastally trapped waves (CTWs) are a major cause of sub-seasonal coastal sea level variability. While they have mostly been studied using numerical models, observational evidence is limited due to the sparse spatial coverage of the tide gauge network and the limitations of satellite altimetry gridded maps, which arise from the interpolation of sparse along-track data. The simultaneous operation of multiple altimetry missions, advancements in processing technologies, the advent of wide-swath altimetry, and the development of new interpolation techniques have the potential to significantly improve the monitoring of CTWs. In this study, we analyze three months of sea level data from satellite altimetry to evaluate the new capabilities for detecting sub-monthly variability, comparing the results to tide gauge data and an ocean model in Eastern Australia, an area known for its dominance of CTWs at these time scales. The results demonstrate that in the study area, the correlation between tide gauges and coastal daily sea level grids from satellite altimetry exceeds 0.5, even when considering time series filtered to capture only sub-monthly variability. CTWs are generally well detected, though discrepancies remain, particularly in terms of amplitude, wavelength and period.

# 1 Introduction

Although long-term sea level variability is dominated by annual and interannual variations, it is known from in-situ data, models, and reanalysis that strong variations occur at submonthly scales, particularly in the shelf seas and coastal zones (Woodworth et al., 2019). One of the main dynamics in these regions is coastally trapped waves (CTWs), which are forced by winds and propagate the energy transmitted from the atmosphere to the ocean (Hughes et al., 2019). Although these phenomena have been reported in many regions of the world (see Aydın and Beşiktepe (2022) for a comprehensive list), a systematic observational-based analysis is made difficult by the coarse spatial resolution of the global tide gauge network (Woodham et al., 2013). However, the literature has mainly relied on these in-situ data and on models to study CTWs, while remotely sensed sea level data from satellite altimeters has been used only to study large-scale phenomena at temporal scales of a month or more (Poli et al., 2022; Polo et al., 2008; Kemgang Ghomsi et al., 2024). The objective of this work is to provide a first assessment of the capability of state-of-the-art sea level maps based on satellite altimetry to detect sub-monthly CTWs. As a testbed, the Eastern Coast of Australia was chosen. This choice is justified by the fact that the region is characterized by a narrow shelf, whose sea level variability is dominated by CTWs on timescales ranging from one day to several months (Maiwa et al., 2010).

Moreover, Woodham et al. (2013) has shown that model data from the Bluelink ocean forecasting system (Schiller et al., 2019) can effectively describe the CTWs and can be validated with the local tide gauge network.

Satellite altimetry is a remote sensing technique based on the analysis of the returned signal sent from a radar, which provides the measurement of the distance between the satellite and the ocean surface (range) (Chelton et al., 2001). The range is then corrected for instrumental, atmospheric, and geophysical effects and subtracted from the orbit altitude to obtain the sea level height (Andersen and Scharroo, 2011). The data collected along the track of each satellite are then referred to a mean sea surface, cross-calibrated, and interpolated in the form of regularly spaced grids (Le Traon et al., 1998). Although in principle sea level maps from satellite altimetry could provide an excellent monitoring system for CTWs, their use has been limited by two main reasons. First, the need for spatio-temporal interpolation and the reduced data quality in the coastal zone may affect the ability to spatially constrain the CTWs, particularly in narrow shelves. Secondly, although these grids are currently released at a daily rate, previous literature has defined their effective temporal resolution to be about one month (Ballarotta et al., 2019). This points more to a worsening of the signal-to-noise ratio at higher frequencies, rather than to a complete absence of a signal. Moreover, the effective resolution strongly depends on the number of satellites used for data collection.

In recent years and months, several innovations have occurred that could play a central role in overcoming these limitations. There have never been so many altimeters in orbit, which means that the amount of data is significantly higher than in previous years (The International Altimetry Team, 2021). The quality of the data in the coastal zone has been significantly improved by the use of new processing techniques and better geophysical corrections (Donlon et al., 2021). The successful launch of the Surface Water and Ocean Topography (SWOT) mission means that, for the first time, two-dimensional snapshots of sea level at an unprecedented spatial resolution can be combined with traditional one-dimensional along-track data (Fu et al., 2024). New interpolation algorithms have been designed to exploit the potential improvements coming from these innovations (Juhl et al., 2025; Passaro and Juhl, 2023; Ballarotta et al., 2025; Beauchamp et al., 2023). In the following sections, their impact and importance in opening up new possibilities for CTWs monitoring are evaluated and discussed.

The outline of the manuscript is as follows: Section 2 presents the data used in the analysis, including the study area. Section 3 describes the methodology applied to analyze the sub-monthly CTWs. Section 4 reports the results obtained, which are then discussed in Section 5, along with concluding remarks and future perspectives.

#### 50 **2 Data**

30

## 2.1 Tide Gauges

Tide gauges (TG from now on) are used in this study to compare the results obtained from other data sources. High-frequency (hourly) data, similar to those distributed by the Global Extreme Sea Level Analysis (GESLA-3, www.gesla.org, Woodworth et al. (2016)), are used. Since the GESLA-3 data available online end in 2022, a specific update was requested, and data for 2023 were obtained through personal communication with Ivan Haigh (University of Southampton, UK) and Ben Hague (Bureau of Meteorology, Australia).

Stations in Eastern Australia within the rectangle visible in Figure 1 were selected. Stations were retained if they ensured a minimum spacing of 50 km between them and were at least 25 km away from a river mouth. Based on data availability and these criteria, the following stations were used: Bermagui, Crookhaven\_Heads, Port\_Macquarie, Coffs\_Harbour, and Brunswick Heads.

To make the sea level retrievals from TGs comparable to other sources of sea level data, the following steps were performed. Firstly, the TG data were detided using a 40-hour LOESS filter. Previous literature has shown this to be the most effective method for reducing tidal variance for periods shorter than two days (Saraceno et al., 2008). Subsequently, the atmospheric contribution to sea level variability was removed by applying the same correction used for altimetry data, the Dynamic Atmosphere Correction from Carrère and Lyard (2003). Finally, the hourly and sub-hourly data collected by each TG were averaged to a daily rate to match the data rate of the altimeter product.

# 2.2 Ocean Model

60

The approach of Woodham et al. (2013) is followed, which studied CTWs in the same region using data from the Bluelink ReANalysis (BRAN) experiment (referred to simply as BLUELINK in the following sections). The latest version, BRAN2020, is used. It simulates the period from 1993 to 2023 using a near-global, eddy-resolving ocean model with a 10-km spatial resolution and daily temporal resolution. Through data assimilation, the model integrates observations of temperature, salinity, and sea surface height (from satellite altimetry) to refine the ocean state. Details of the model and an analysis of its performance can be found in (Chamberlain et al., 2021).

# 2.3 Altimetry

- 75 The altimetry data used in this study come from two distinct products:
  - CMEMS: the operational daily sea level grids distributed by the Copernicus Marine Service, based solely on nadir altimeters.
  - MIOST<sub>SWOT+nadirs</sub>: an experimental product distributed by AVISO that integrates nadir altimetry with swath data from the SWOT mission.
- All altimetry data used are provided at a daily rate. The latest generation, named DT2024 and available since November 2024 in the Copernicus Marine Service (CMEMS), applies for the first time a different approach compared to the optimal interpolation adopted in previous versions (e.g., Le Traon et al. (1998); Ducet et al. (2000); Pujol et al. (2016)): the Multiscale Inversion of Ocean Surface Topography (MIOST) technique (Ubelmann et al., 2021, 2022).

The MIOST technique solves the mapping problem by integrating all components within a reduced space and utilizing a preconditioned conjugate gradient method (Ubelmann et al., 2021). This iterative process continuously refines the solution until convergence. Once the final reduced solution is obtained, it is projected back onto the full spatial grid using wavelet-based transformations specific to each component.

The MIOST technique is also applied to an experimental product distributed by AVISO (called MIOST<sub>SWOT+nadirs</sub> in this study), representing the first attempt to combine traditional Level-3 along-track sea level anomalies with the two-dimensional Level-3 product from the SWOT mission (Dibarboure et al., 2025). Besides the integration of the swath-altimeter data, minor methodological differences include a different selection of the components of the wavelet-based transformations (M. Ballarotta, personal communication).

Three months of data (from September 2023 to November 2023) are considered during the science phase of SWOT. In this period, both CMEMS and MIOST<sub>SWOT+nadirs</sub> blend along-track measurements from the following nadir altimeters: SAR-AL/AltiKa, Cryosat-2, HaiYang-2B, Jason-3, Sentinel-3A, Sentinel-3B, Sentinel-6A, and SWOT nadir.

# 3 Methods

# 3.1 Filtering

For each latitude-longitude point of all the data sources listed above, a time series is produced and stored at a daily rate, which is referred to as unfiltered sea level. Subsequently, the time series is band-pass filtered using a Butterworth bandpass filter, filtering out components longer than 29 days (0.15 cycles per day in frequency) and shorter than 7 days (0.035 cycles per day in frequency).

# 3.2 Lag-correlation

A correlation analysis is performed to study the consistency of the dataset with the ground truth represented by the TG data.

The computation is based on the Pearson correlation coefficient to measure the linear relationship between time series x and y:

$$r = \frac{\sum_{i=1}^{n} (x_i - \bar{x})(y_i - \bar{y})}{\sqrt{\sum_{i=1}^{n} (x_i - \bar{x})^2} \sqrt{\sum_{i=1}^{n} (y_i - \bar{y})^2}}$$
(1)

where  $x_i$  and  $y_i$  are the individual sample points of the two time series,  $\bar{x}$  and  $\bar{y}$  are the mean values of x and y, respectively, and n is the number of observations.

To observe the along-shore coherency of intervals, the lag-correlation is also computed. In this case, the correlation is computed between  $x_t$  and a time-shifted version of  $y_t$ , denoted as  $y_{t+\tau}$ , where  $\tau$  represents the lag (in days):

$$r(\tau) = \frac{\sum_{i=1}^{n-\tau} (x_i - \bar{x})(y_{i+\tau} - \bar{y}_{\tau})}{\sqrt{\sum_{i=1}^{n-\tau} (x_i - \bar{x})^2} \sqrt{\sum_{i=1}^{n-\tau} (y_{i+\tau} - \bar{y}_{\tau})^2}}$$
(2)

#### 3.3 Hovmöller diagrams and Phase-speed computation

The average propagation speed of the CTWs can be estimated from the Hovmöller diagrams. These diagrams are obtained by representing the temporal evolution of sea level anomalies along equally spaced points along the coastline, which are associated

with the closest grid point of each dataset. The propagation speed of the observed contours is then computed based on the slopes that can be derived using image processing methods. The method based on the Radon transform, as described in De-Leon and Paldor (2017), is followed. First, the Hovmöller diagrams are normalized to the range 0–1. The Radon transform works by considering straight lines in the 2D field, which are rotated by an angle  $\theta$  and shifted by a distance s. It is defined as:

$$R_f(s,\theta) = \int_{-\infty}^{\infty} f(s\cos\theta - t\sin\theta, s\sin\theta + t\cos\theta) dt,$$
(3)

where  $R_f(s,\theta)$  is the Radon transform of f(x,y), s is the perpendicular distance of the line from the origin,  $\theta$  is the angle of the line relative to the ordinate, and t is a parameter along the line L, integrating over all points on that line. The transform sums the squares of the 2D field along each straight line to quantify the energy of the features aligned at each angle:

$$E(\theta) = \int_{-\infty}^{\infty} R_f^2(s,\theta) \, ds. \tag{4}$$

The best estimate of the westward propagation speed is then the tangent of the angle at which the sum of the squares reaches its maximum:

$$v = \tan(\theta_{\text{max}}),\tag{5}$$

where v represents the westward propagation speed.

#### 3.4 Complex Empirical Orthogonal Functions Analysis

To determine whether CTWs represent a dominant pattern of sub-monthly coastal sea level variability, a Complex Empirical Orthogonal Function (CEOF) analysis is performed using all time series of filtered sea level within the shelf domain, characterized by bathymetry shallower than 500 meters.

The CEOF analysis is a well-known technique to identify variability patterns in time and space and has also been used for this purpose in previous studies of CTWs such as Woodham et al. (2013). In this analysis the data are arranged as a complex matrix, where the real part is the original data, and the imaginary part is the Hilbert-transformed data. The coastal time series can be represented as a matrix  $\mathbf{X}$ , where each row corresponds to a spatial location and each column corresponds to a time step. An anomaly matrix  $\mathbf{X}'$  is then computed by subtracting the mean at each spatial location to remove the temporal trend. To identify the dominant spatial patterns, the complex anomaly matrix using the Hilbert transform is defined:

$$\mathbf{X}_c = \mathbf{X}' + i \cdot \mathcal{H}(\mathbf{X}') \tag{6}$$

where  $\mathcal{H}(\mathbf{X}')$  is the Hilbert transform of  $\mathbf{X}'$ , and i is the imaginary unit.

The complex covariance matrix is computed as:

$$\mathbf{C} = \frac{1}{N} \mathbf{X}' \mathbf{X}'^{\mathrm{T}}. \tag{7}$$

where N is the number of time steps. The CEOF modes are obtained by solving the eigenvalue problem:

$$\mathbf{C}\mathbf{e}_{k} = \lambda_{k}\,\mathbf{e}_{k} \tag{8}$$

where  $e_k$  are the complex eigenvectors (the CEOF spatial modes), and  $\lambda_k$  are the eigenvalues, representing the variance explained by each mode. Finally, the Principal Component (PC) time series describes how the CEOF modes vary over time:

$$PC_k = \mathbf{e}_k^{\mathrm{T}} \mathbf{X}'. \tag{9}$$

# 4 Results



## 4.1 1D-Validation against tide gauges

In Figure 2, an example of the time series from all data sources closest to one TG is presented, which can be qualitatively discussed before a quantification of the correlation is provided. The time series are shown in both their unfiltered and filtered versions. A visual inspection of the TG data reveals oscillations with a period of approximately 10 days and amplitudes ranging from 5 to 10 cm. The strongest one is found around day 50 and corresponds to a maximum in the SLA. For the purposes of this section, it is assumed that the observed oscillations represent the signature of CTWs, which are known to dominate sub-monthly variability in the study region (see Introduction). This assumption will be further examined in the following sections.

The BLUELINK time series matches very well the TG data, capturing all the CTWs and showing a good match in terms of phase as well. Only the peak of the strongest event is slightly underestimated. The CMEMS data fail to capture most of the sub-monthly oscillations and underestimate the amplitude peak. The MIOST<sub>SWOT+nadirs</sub> data show much better agreement with the TG and BLUELINK, corresponding to at least five of the CTWs correctly observed, with amplitude and phase similar to the BLUELINK and TG estimations. However, events that almost superimpose cannot be correctly distinguished, as clearly seen between days 40 and 60.

To quantify the agreement, the correlation of BLUELINK, MIOST<sub>SWOT+nadirs</sub>, and CMEMS against each TG for the unfiltered and filtered time series was computed and the results are reported in Figure 3A and B. BLUELINK has the best score, showing higher agreement with the TGs at every station for both the unfiltered and filtered time series. Notably, the correlation coefficient is never lower than 0.7, and there is no strong sign of worse performance in the filtered version. This confirms the findings of Woodham et al. (2013) and allows us to use the reanalysis as a validation of the altimetry-based dataset in this

study, in addition to the TGs, which have the obvious limitation of providing only point-wise time series at a limited number of locations and therefore contain limited spatial information.

In the unfiltered time series, MIOST<sub>SWOT+nadirs</sub> performs significantly better than CMEMS, with a correlation coefficient never lower than 0.6. Most notably, while a drop in correlation is observed for both altimetry data sources when considering the filtered time series, a strong improvement is seen in MIOST<sub>SWOT+nadirs</sub> compared to CMEMS. The correlation between the filtered MIOST<sub>SWOT+nadirs</sub> and TGs is higher than 0.5 in four out of five stations, while for CMEMS the correlation exceeds 0.5 only in one station. The largest drop in correlation between unfiltered and filtered signals in the altimetry dataset occurs in Bermagui, the station closest to the shelf break, which in that region is located 30-40 km from the coast. The drop in correlation at high frequencies is therefore likely due to the influence of open ocean processes in the estimation of the coastal grid points.

Figure 3C and D presents the spectral analysis of the unfiltered and filtered time series, respectively, performed using the Power Spectral Density (PSD) estimated with Welch's method. For this analysis, a Hann window and a segment length of 45 data points were used, with the PSD averaged across the five selected tide gauges. The shaded areas represent the uncertainty, computed as the standard deviation of the PSD across these gauges at each period. The tide gauges and BLUELINK exhibit a robust energetic local maximum at around 10 days, while the energetic content of the altimetry datasets is shifted toward longer periods. All datasets exhibit significant energy at sub-monthly periods, although a clear characterisation is hindered by the shortness of the time series.

## 4.2 2D-Validation against tide gauges




Besides their temporal signature, we expect the sub-monthly CTWs identified in the previous section to also have a spatial signature along the shelf. The sea level oscillation caused by the CTWs can typically be observed by computing lag correlations. Moreover, Woodham et al. (2013) and our results give us confidence that the BLUELINK data can serve as a reference for the altimetry dataset. Using the filtered time series, we therefore compute the correlation coefficient between each grid point and the Bermagui TG at different lags of 0, 2, and 4 days. The results are shown in Figure 4. Bermagui is chosen because it is the southernmost location in the domain, in order to highlight, through lag-correlation, the spatial footprint of coastally trapped waves traveling northward. However, the same statistics have been produced for every TG station, and the corresponding figures can be found in the Appendix (Figures B1 to B4).

Taking BLUELINK as the reference, it is possible to observe how the correlation pattern is very well constrained within the shelf, which is defined by the detached line showing the bathymetric contour at -500m. The locations to the north of -31 degrees S on the shelf are anticorrelated with Bermagui at Lag 0, while the same region shows correlations well above 0.5 with a lag of 4 days. In MIOST<sub>SWOT+nadirs</sub>, the same pattern as in BLUELINK can be traced, although the average correlation with the TG is generally lower than in BLUELINK, and the areas of high lag-correlation are slightly more spread outside the shelf. CMEMS also shows a similar pattern, but the level of correlation, especially at Lag 2 and Lag 4, is much lower than in MIOST<sub>SWOT+nadirs</sub> and therefore less distinguishable from the random patterns in the open ocean. Observing the correlations at Lag 0, it is possible to better understand the drop in performance for Bermagui seen in Figure 3: this is due to an eddy-shaped

**Figure 1.** Region of study. The parallelogram shows the region in which sea level data from reanalysis and remote sensing is acquired. The tide gauge stations used in the work are also shown.

**Figure 2.** Daily sea level data from CMEMS, MIOST<sub>SWOT+nadirs</sub>, BLUELINK and the Coffs\_Harbour tide gauge. The upper plot shows the full signals, while the lower plot is obtained by filtering out components longer than 29 days (0.15 cycles per day in frequency) and shorter than 7 days.

uncorrelated feature that is not visible in the model and that MIOST<sub>SWOT+nadirs</sub> and CMEMS localize on both sides of the shelf break.

**Figure 3.** Upper panels: Correlation of CMEMS, MIOST<sub>SWOT+nadirs</sub> and BLUELINK time series with the TG data at the closest grid point to each in-situ station. The plot shows the correlation considering unfiltered (A) and filtered (B) time series. Lower panels: The averaged PSD of the same time series as above in their unfiltered (C) and filtered (D) version. The shaded areas represent the standard deviation across these gauges at each period.

## 4.3 Phase speed and EOF analysis



After subdividing the coastline into equally spaced points, we analyze the Hovmöller diagram of the filtered signals in the three datasets and compute the average phase speed of the CTWs based on the Radon transform. The results are shown in Figure 5. The pattern of the CTWs traveling anticlockwise along the Australian coastline is very clearly modeled by BLUELINK, but can also be recognized in MIOST<sub>SWOT+nadirs</sub> and, partially, in CMEMS. The filtered altimetry time series in MIOST<sub>SWOT+nadirs</sub> are good enough that it is possible to measure exactly the same phase speed of the CTWs as in BLUELINK (4.01 m/s), while the phase speed estimated for CMEMS is still very close (4.33 m/s). This is a striking result, considering that we are using the time series closest to the coastline, meaning the points for which the quality of the altimetry data is supposed to be the lowest. As the next step, we analyze the results of the CEOF analysis applied to the filtered signal. The reconstructed signals from the first CEOF are shown as Hovmöller diagrams in Figure 6 using the same coastal locations as previously. As expected from

previous studies, BLUELINK shows a dominant first mode 71% of the total variance) characterized by an oscillating pattern

**Figure 4.** Correlation coefficient (*r*) of filtered CMEMS, MIOST<sub>SWOT+nadirs</sub> and BLUELINK time series at each grid point against the filtered TG data of the station Bermagui (black triangle). Each row corresponds to the lag correlation at day 0, 2, 4. The bathymetry contour of -500m is shown to identify the shelf break. Locations where the correlation is not statistically significant are marked with black diagonal stripes.

typical of CTWs. A similar first mode is seen with different degrees of similarity also in MIOST<sub>SWOT+nadirs</sub> and CMEMS, although the degree of variance explained is less in the altimetry dataset (55% for MIOST<sub>SWOT+nadirs</sub>, 52% for CMEMS).



The use of CEOF analysis enables the examination of the signal's phase in both spatial and temporal dimensions. Both MIOST<sub>SWOT+nadirs</sub> and BLUELINK exhibit a clear linear phase progression along the coastline (6, lower panels), characteristic of traveling waves, whereas this pattern appears noisier in CMEMS. By analyzing the slope of the spatial phase and the corresponding spatial distance, the spatial wavelength of the signal can be estimated. The mean spatial wavelength is approximately 2365 km for BLUELINK and 3142 km for MIOST<sub>SWOT+nadirs</sub>. These values are challenging to validate: Woodham et al. (2013) reported a mean spatial wavelength of 4000 km in the same domain for 2009 using BLUELINK, while in-situ campaigns from the 1980s cited in the same study reported wavelengths around 2500 km. For the purposes of this study, it is important that

**Figure 5.** Hovmöller diagrams of CMEMS, MIOST<sub>SWOT+nadirs</sub> and BLUELINK filtered time series. Each column represents the sea level anomaly time series at a coastal location within the parallelogram shown in Figure 1, progressing from the southwesternmost point northward along the coast. The dotted lines correspond to the best estimate of the westward propagation speed (see Section 3.3). The x-axis represents the distance from the southwesternmost point along the coastline.

the average spatial wavelength is within a realistic range and that the spatial phase decreases linearly in both BLUELINK and MIOST<sub>SWOT+nadirs</sub>. In contrast, the average wavelength of 1211 km derived from CMEMS appears unrealistic due to the irregularity of its spatial phase slope. The temporal phase of the leading mode increases linearly across all three datasets, indicating a sinusoidal oscillation with a dominant period. The average period of the first mode is approximately 10 days for BLUELINK, 14 days for MIOST<sub>SWOT+nadirs</sub>, and 18 days for CMEMS.

A confirmation of this evaluation is evident in the temporal and spatial evolution of the reconstructed signal from the first CEOF, presented in Figure 7 for three days during the strongest CTW event observed in the second half of October 2023. MIOST<sub>SWOT+nadirs</sub> and BLUELINK exhibit a consistent pattern of anticlockwise oscillating amplitudes, although in BLUELINK the decreasing amplitude of the CTW from onshore toward the shelf is more clearly visible. BLUELINK displays the highest CTW amplitudes, while these are strongly damped in CMEMS. Similar conclusions can be drawn from the filtered signal shown in Figure C1.

## 5 Discussion and conclusion





While gridded sea level maps have often been used in the coastal zone for comparison with tide gauges (e.g., Wöppelmann and Marcos (2016); Oelsmann et al. (2024)), they are usually evaluated as monthly averages or to observe annual, interannual, and long-term trends.

Indeed, Passaro et al. (2023) and Juhl et al. (2024) have shown in the North Sea and in the Patagonian Shelf that the coherence between TGs and daily sea level maps consistently drops for periods below 30 days, which is in agreement with the effective temporal resolution reported by Ballarotta et al. (2019). However, we notice a strong improvement in the correlation of the filtered signal when using the latest version of the CMEMS product (DT2024) compared to the previous one (DT2021, Sánchez-Román et al. (2023)), which still used the optimal interpolation and the same altimetry missions. This can be seen in Figure A1,

**Figure 6.** Upper plots: Hovmöller diagrams showing the amplitude of the reconstructed signal from the first CEOF for CMEMS, MIOST<sub>SWOT+nadirs</sub> and BLUELINK. Each column represents the reconstructed sea level anomaly time series at a coastal location within the parallelogram shown in Figure 1, progressing from the southwesternmost point northward along the coast. Lower plots: Spatial (left) and temporal (right) phase of the first CEOF.

where we show the results obtained with the same analysis but using DT2021 instead of DT2024. The enhanced capabilities in observing sub-monthly variability are therefore a consequence of two factors: the adoption of a different interpolation scheme based on wavelet transformation and the integration of SWOT data.

The availability of the same interpolation scheme with and without the use of wide-swath altimetry allows us to observe the impact that the latter is having for this particular application. This impact was recently analyzed by Ballarotta et al. (2025), who concluded that the benefits are limited to an improvement of 10 km for the spatial resolution on average. However, their analysis did not consider the improvements in the observation of the temporal scales, and more specifically the coastal zone. Here, we provide the first proof that, at least for shelves wider than 40 km, the combination of the MIOST interpolation with the integration of the wide-swath data is able to detect the CTWs in terms of average phase speed (Figure 5) and spatial pattern (CEOF in Figure 6), although the accuracy in terms of amplitude, wavelength and period of the CTWs is still subobptimal.

This gap in accuracy could be significantly reduced in the next few years with the launch of the Copernicus Sentinel-3 New Generation (S3-NG) Topography mission, which, according to current plans, should consist of two wide-swath altimetry missions. This would compensate for the disparity between very high spatial resolution and relatively long repeat cycles. The causes of the remaining gap should also be investigated, with a focus on the impact of the Dynamic Atmosphere Correction (DAC, Carrère and Lyard (2003)). This correction is currently applied to the along-track measurements that are then interpolated into the gridded products. Besides the low-frequency static effects due to the inverse barotropic response, it is based on a

**Figure 7.** Temporal and spatial evolution of the reconstructed signal from the first CEOF mode for CMEMS, MIOST<sub>SWOT+nadirs</sub>, and BLUELINK, shown for three selected days during the most intense CTW event observed in the second half of October 2023.

simulation of the wind-forced barotropic motions by means of the MOG2D-G ocean model for periods shorter than 20 days. The objective of the correction is to remove the signal that would be aliased at lower frequencies on the single along-track records due to their long repeat cycles (never shorter than 10 days). In our region of study, CTWs could also be caused by a direct response of the ocean surface to the changing wind field (Woodham et al., 2013), and therefore their observations could be affected by the application of the DAC.


This study suggests that, especially with the increasing availability of MIOST<sub>SWOT+nadirs</sub> data during the SWOT era, it is possible to establish an altimetry-based monitoring system for CTWs using filtered time series focused on a set of grid points

as "virtual stations." This approach would complement and strengthen studies currently based on sparse in-situ stations (when 265

available) and numerical simulations. Such studies investigate how CTWs control upwelling and productivity in coastal areas,

for example, off Peru (Echevin et al., 2014) and in South West Africa (Bachèlery et al., 2020), where CTWs account for 70%

of the high-frequency variability.

Code and data availability. The complete code used to generate the statistics and plots in this work is publicly available from: https://github.

com/ne62rut/coastal\_trapped\_waves

The tide gauge data are an extension of the GESLA dataset (https://www.gesla.org/). They were obtained courtesy of Ivan Haigh (National

Oceanography Centre, University of Southampton). The Bluelink Ocean Reanalysis - BRAN2020 (BLUELINK) data were downloaded from

in August 2024 from https://geonetwork.nci.org.au/. The CMEMS data is the SEALEVEL\_GLO\_PHY\_L4\_MY\_008\_047 downloaded

from https://marine.copernicus.eu/ downloaded first in August 2024 (Version DT2021) and then in February 2025 (Version DT2024). The

MIOST<sub>SWOT+nadirs</sub> data is the "Experimental multimission gridded L4 sea level heights and velocities with SWOT" product available from

https://doi.org/10.24400/527896/A01-2024.007, downloaded in July 2024.


Appendix A: 1D-Validation against tide gauges using DT2021

Appendix B: 2D-Validation against tide gauges

**Appendix C: Single CTW event** 

**Figure A1.** Correlation of CMEMS DT2021, MIOST<sub>SWOT+nadirs</sub> and BLUELINK time series with the TG data at the closest grid point to each in-situ station. The plot shows the correlation considering unfiltered (upper panel) and filtered (lower panel) time series. CMEMS DT2021 is the previous version of the dataset used Figure A1, which was the standard until November 2024.

Figure B1. Correlation coefficient (r) of filtered CMEMS, MIOST<sub>SWOT+nadirs</sub> and BLUELINK time series at each grid point against the filtered TG data of the station Brunswick\_Heads (black triangle). Each row corresponds to the lag correlation at day 0, 2, 4. The bathymetry contour of -500m is shown to identify the shelf break.

Figure B2. Correlation coefficient (r) of filtered CMEMS, MIOST<sub>SWOT+nadirs</sub> and BLUELINK time series at each grid point against the filtered TG data of the station Coffs\_Harbour ((black triangle). Each row corresponds to the lag correlation at day 0, 2, 4. The bathymetry contour of -500m is shown to identify the shelf break.

Figure B3. Correlation coefficient (r) of filtered CMEMS, MIOST<sub>SWOT+nadirs</sub> and BLUELINK time series at each grid point against the filtered TG data of the station Crookhaven\_Heads (black triangle). Each row corresponds to the lag correlation at day 0, 2, 4. The bathymetry contour of -500m is shown to identify the shelf break.

Figure B4. Correlation coefficient (r) of filtered CMEMS, MIOST<sub>SWOT+nadirs</sub> and BLUELINK time series at each grid point against the filtered TG data of the station Port\_Macquarie (black triangle). Each row corresponds to the lag correlation at day 0, 2, 4. The bathymetry contour of -500m is shown to identify the shelf break.

**Figure C1.** Temporal and spatial evolution of the filtered signal from CMEMS, MIOST<sub>SWOT+nadirs</sub>, and BLUELINK, shown for three selected days during the most intense CTW event observed in the second half of October 2023.

Author contributions. M.P. conducted the research, performed the data analysis, and wrote the manuscript.

Competing interests. No competing interests are present


Acknowledgements. Thanks to Ivan Haigh (National Oceanography Centre, University of Southampton, UK) and Ben Hague (Bureau of Meteorology, Australia) for the help in providing tide gauge data. Thanks to Maxime Ballarotta (Collecte Localisation Satellites, France) for the assistance concerning MIOST.

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
