# Peer review of "Can satellite altimetry observe coastally trapped waves on sub-monthly timescales?"

_EGUsphere, 2025_

## Referee Comment (RC1)

**Can satellite altimetry observe coastally trapped waves on sub-monthly timescales?**

Author(s): Marcello Passaro

MS No.: egusphere-2025-809

**Does the paper address relevant scientific questions within the scope of OS?**

Yes, the manuscript addresses the capacity of new altimetric products to detect CTW signals at periods shorter than 29 days, a challenging task for existing altimetry products. Observing coastal sea level is crucial for understanding variability along the coast.

**Does the paper present novel concepts, ideas, tools, or data?**

Yes, the study used the new MIOST dataset, which includes SWOT altimetric satellite measurements.

**Are substantial conclusions reached?**

Yes, the results indicate that including SWOT data enhances the ability to capture coastal signals.

**Are the scientific methods and assumptions valid and clearly outlined?**

The scientific methods are clearly described. However, further clarification and adjustments may improve certain analyses.

**Are the results sufficient to support the interpretations and conclusions?**

More details are needed regarding the statistical significance of results. An  analyses on CTWs associated spatial patterns would be nice to show the improvement of the inclusion of SWOT in altimetric DATA in the representation of fine scale cross-shore structures .

**Is the description of experiments and calculations sufficiently complete and precise to allow their reproduction by fellow scientists (traceability of results)?**

The methodology is well described.

**Do the authors give proper credit to related work and clearly indicate their own new/original contribution?**

Yes, the author appropriately cite related work and clearly distinguish their contributions.

**Does the title clearly reflect the contents of the paper?**

Yes, the title accurately reflects the content of the paper.

**Does the abstract provide a concise and complete summary?**

Yes overall the abstract provides a concise summary but could include the different used altimetric products and mentioned SWOT.

**Is the overall presentation well structured and clear?**

The manuscript is well-structured but requires additional clarification in some sections.

**Is the language fluent and precise?**

The language is mostly fluent and precise, but minor improvements could enhance clarity.

**Are mathematical formulae, symbols, abbreviations, and units correctly defined and used?**

Yes, mathematical expressions, symbols, abbreviations, and units are correctly used.

**Should any parts of the paper (text, formulae, figures, tables) be clarified, reduced, combined, or eliminated?**

I Answer this in the specific comments.

**Are the number and quality of references appropriate?**

Yes , it is appropriate.

**Is the amount and quality of supplementary material appropriate?**

Yes , it is appropriate.

The manuscript evaluates the capacity of new altimetric products which include new satellites especially SWOT which measures sea level along 2D swath, and new processing techniques to capture Coastal trapped waves (CTWs) signal at submontly time-scales. The author describes the methodology and applies statistical analysis ( correlations and Empirical Orthogonal function decomposition ) to assess the different products and found that the submonthly signals are overall well detected by altimetry. This manuscript presents a valuable analysis of new altimetric products for capturing CTWs. However, several aspects require further clarification and refinement. Hereafter the specific comments. I think addressing these points will strengthen the manuscript and improve its impact.

**Introduction**

- To improve clarity, I suggest reorganizing the introduction into two distinct paragraphs:
  A discussion on CTWs as the primary study objective, emphasizing the importance of accurately representing CTWs and a description of the different altimetric products (MIOST with and without SWOT) to avoid repetition and potential confusion.
- Additionally, it would be helpful to include an outline of the paper at the end of the introduction.

**Data**

*Tide Gauges :*

How many tide gauges were excluded?
*Altimetry :*

Introducing the different products at the beginning and specifying their names would improve clarity.
*Lines 82–85:*

Does the inclusion of SWOT in the MIOST product significantly affect the dataset compared to DT2024 in the study region during the scientific phase? Specifically, does it affect spatial patterns ? To address this in the results,  it would be helpfull to add  snapshots of a CTW event.

**Methods**

*Section 3.2 :* This paragraph may not be necessary, as the method is well-known.
*Section 3.3 :* This section may also be unnecessary.

**Results and Discussion**

*Time series :*

I think that conducting a spectral analysis of the unfiltered time series would be useful to quantitatively compare the different products and support the affirmation in line 145-146. Figure 2 and 3 could be also combined.

*Correlations ( Figure 4) :*

- Are the correlations shown statistically significant, particularly offshore? Some correlation values on the shelf appear to be of the same magnitude as those offshore. Applying a mask could help focus on shelf values.
- Could the lag provide an estimated CTW period? Is this period consistent with those observed in the Hovmöller diagrams?
- Additionally, the continuous colorbar makes it difficult to discern correlation values.

*Section 4.3 :*

- Can you indicate the track used for the Hovmöller diagram in Figure 1 for more clarity.
Does selecting a more offshore track affect CMES and MIOST results (Line 189)?

- Lines 186–187 : that can be move to the figure caption.

*Phase Speed Computation :*

- The computed phase speed appears to be underestimated since the dashed line does not cross SLA maxima for BLUELINK, and the maxima are difficult to identify in CMES data.I was wondering how does the Radon transform method compute phase speed in the absence of a clear propagation pattern?
- Is the phase speed computed only for the first propagation, and does it remain consistent with subsequent propagations? For example, the CTWs propagating between 10/15/2023 and 11/01/2023 seem better captured by MIOST and CMES. Do these waves share the same phase speed?
- I suggest to compute SLA lagged correlations along the track from a reference point (e.g., Bermagui) . Then, estimate phase speed using a distance vs. lagged correlation plot.

*EOF Analyses:*

- Using EOFs to study CTWs might introduce biases, as the same propagation pattern could be projected onto different modes of variability. To better capture propagative patterns, consider using complex EOFs.
- How does the other EOFs modes appear?
- Consider presenting EOFs results in a "Hovmöller aesthetic," with distance on the x-axis, days on the y-axis, and EOF amplitude in color. This could help reduce figure size and enhance clarity.
- Merging Figures 6 and 7 may also be beneficial.

*CTW characterizations:*

I think the paper lacks a clear description of the CTWs observed in terms of period and spatial patterns. You could apply a complex EOF analysis on offshore-masked maps to estimate wave wavelengths an phase. Or Adding snapshots would illustrate how different products reproduce spatial patterns associated with CTWs.

---

## Referee Comment (RC2)

Review of
**Can satellite altimetry observe coastally trapped waves on sub-monthly timescales?**
Marcello Passaro

**General comments**
The manuscript focuses on the potential of improved altimetry products to detect coastally trapped waves (CTW). So far, those waves were identified using tide gauge data that is very accurate with a high temporal resolution but sparsely spaced along the worlds coastlines. The manuscript successfully demonstrates that new products are able to show how CTWs propagate anticlockwise along the eastern Australian coast.
The author uses a combination of standard (correlations, EOFs) and (to me) less known techniques (image processing) to arrive at their conclusion. I find the argumentation convincing but I think the presentation of the results could be improved (see below).

**Specific comments**
Line 68: Remove parentheses for Woodham et al.
Figure 1: I found the green rectangle a bit hard to see. Consider a different color and thicker lines.
Line 68: Can you say something about the temporal resolution of this dataset, and how this compares to the other datasets used in this study?
Line 71: what kind of sea surface height data is assimilated? Remote- sensing data I assume?
Line 86: To be clear: The major difference between CMEMS and MIOST is the inclusion of the swath-altimeter data in MIOST? Both, MIOST and CMEMS use the MIOST technique to solve the mapping problem? I found that a bit confusing and would recomment choosing a different short name for what is now the MIOST product.

Line 95: Here you assume that CTWs have periods between 7 and 29 days? Can you justify this? Have most of the CTWs observed so far fallen in that range? (Aydın and Beşiktepe, 2022, state that CTWs typicall fall into the 8-16d range and I'm wondering why you extend your range to 29 days.)
Equation 7: T is the number of time steps but also denotes the transposed matrix X', right? Maybe choose another letter for the number of time steps (N?) to avoid confusion.

Line 146: "The TG clearly shows…": is this from visual inspection only or is there more evidence?
Figure 2: Legends: A bit pedantic but could you show the TG first or last in the legend? As it is the ground truth. Also, one legend might be enough – the filtered/unfiltered can be in the titles for the two panels. The time step is days?
Line 148: Have you already shown that the variability seen in the data is due to CTWs? Maybe based on previous literature? Or is it only an assumption at this point? Could the oscillations be caused by something else?
Line 172-173: In the previous paragraph you identified Bermagui as the TG with the lowest correlation of the filtered time series with the altimetry data. It seems a bit unfair using that one for the 2D validation. I understand that you choose the southernmost location and that it shouldn't matter as the shelf gets wider northward. But could you still justify that choice?
Figure 5: For better readability consider giving the distance from first point in kilometers on the x-axis. Also, you could have a marker on the x-axis to show the approximate location of the tide gauges such that the Hovmöller diagrams are easier to relate to the maps in Figure 4. In addition, for consistency, use the same labels for the time (y-) axis as in the other figures, i.e. time steps (which are days, I presume?) instead of actual dates.

Line 193: The EOF analysis is very informative but I think it needs to be explained better, particularly the spatial EOFs. You can compare them to the correlation maps in Figure 4 at lag 0 – there, you clearly see a see-saw on the shelf which is consistent with the first spatial EOF (of Bluelink, at least). Also, the figures need to be improved:

Figure 6:
- Shouldn't either the EOF amplitude or the PC amplitude have a unit (m)? That would be useful and would make the results more physical.
- upper panel: what is the location index and how does it relate to the distance in Figure 5?
- lower panel: time steps is days?

Figure 7: The information to be conveyed here is interesting but I wonder if you could do it differently? Instead of showing one panel for each day with all three datasets how about showing only three panels, one for each dataset. In the panels the EOFs could be shown for each time step (using different shades of blue/red/green) so that the anticlockwise rotation becomes very clear.
- as above, what is the location index and how does it relate to the distance in Figure 5?
- I think a unit is missing here.

---

## Author Comment (AC1)

The manuscript evaluates the capacity of new altimetric products which include new satellites especially SWOT which measures sea level along 2D swath, and new processing techniques to capture Coastal trapped waves (CTWs) signal at submontly time-scales. The author describes the methodology and applies statistical analysis ( correlations and Empirical Orthogonal function decomposition ) to assess the different products and found that the submonthly signals are overall well detected by altimetry. This manuscript presents a valuable analysis of new altimetric products for capturing CTWs. However, several aspects require further clarification and refinement. Hereafter the specific comments. I think addressing these points will strengthen the manuscript and improve its impact.

I would like to thank the reviewer for the constructive feedback. Below, I provide a point-by-point response to the comments. The revised sections in the manuscript are highlighted in red.

Note: While this review retains the naming convention used in the original submission to facilitate the review process, the new manuscript adopts the suggestion of the other reviewer by renaming the MIOST product as "MIOST$_{SWOT+nadir}$", which is the same convention used in the following, related paper:

Ballarotta, M., Ubelmann, C., Bellemin-Laponnaz, V., Le Guillou, F., Meda, G., Anadon, C., Laloue, A., Delepoulle, A., Faugère, Y., Pujol, M.-I., Fablet, R., and Dibarboure, G.: Integrating wide-swath altimetry data into Level-4 multi-mission maps, Ocean Sci., 21, 63–80, https://doi.org/10.5194/os-21-63-2025, 2025.

Introduction

- To improve clarity, I suggest reorganizing the introduction into two distinct paragraphs: A discussion on CTWs as the primary study objective, emphasizing the importance of accurately representing CTWs and a description of the different altimetric products (MIOST with and without SWOT) to avoid repetition and potential confusion.

I have taken the reviewer's suggestions into consideration and have reorganized the introduction so that it begins with a discussion on CTWs, followed by a discussion on satellite altimetry. While the paragraphs are distinct, I have not added separate subtitles, as this is not in line with the journal's style. A discussion on the importance of accurately representing CTWs was already included in the conclusions, and I prefer to keep it there. The specific characteristics of the different altimetric products (MIOST with and without SWOT) are addressed in the Data section, where I believe they are most appropriately placed. I believe this new structure of the introduction helps avoid repetition and improves clarity.

- Additionally, it would be helpful to include an outline of the paper at the end of the introduction.

An outline has been added

Data

Tide Gauges : How many tide gauges were excluded?

There were 11 records in the selected region. 3 were excluded to ensure a minimum distance of 50 km between the records. A further 3 were excluded to avoid the proximity of river mouths.

Altimetry :

Introducing the different products at the beginning and specifying their names would improve clarity.

We have applied this suggestion in the new version

Lines 82–85: Does the inclusion of SWOT in the MIOST product significantly affect the dataset compared to DT2024 in the study region during the scientific phase? Specifically, does it affect spatial patterns ? To address this in the results, it would be helpfull to add snapshots of a CTW event.

Combining this suggestion with similar ones later in the review, we have added and commented a new figure showing the snapshots of a CTW event as seen in the CEOF reconstruction based on the different dataset

Methods

Section 3.2 : This paragraph may not be necessary, as the method is well-known. Section 3.3 : This section may also be unnecessary.

I agree with the reviewer that the methods are well known to many readers. However, this may not be the case for everyone. For instance, the second reviewer appears to be less familiar with the Radon transform, which is an image processing technique. Therefore, I would prefer to retain the two paragraphs to enhance clarity and ensure replicability.

Results and Discussion

Time series : I think that conducting a spectral analysis of the unfiltered time series would be useful to quantitatively compare the different products and support the affirmation in line 145-146. Figure 2 and 3 could be also combined.

This is a very good suggestion, I have added the spectral analysis and matched the latter with Figure 3. For this reason, in order not to overload a single Figure, I have left

Figure 2 untouched. The corresponding description of the findings concerning periods and amplitudes has also been amended.

Correlations ( Figure 4) :

- Are the correlations shown statistically significant, particularly offshore? Some correlation values on the shelf appear to be of the same magnitude as those offshore. Applying a mask could help focus on shelf values.

In the new version, a mask using black diagonal stripes is applied to all figures showing correlation within the domain, including those in the appendix, to mark non-statistically significant correlations.

- Could the lag provide an estimated CTW period? Is this period consistent with those observed in the Hovmöller diagrams?

In the new version, the period of the CTW is analyzed using two newly implemented strategies suggested by the reviewer: the complex empirical orthogonal function (CEOF) and the fast Fourier transform (FFT). In my opinion, these methods provide sufficient redundancy.

- Additionally, the continuous colorbar makes it difficult to discern correlation values.

I now use a discrete colorbar

Section 4.3 :

- Can you indicate the track used for the Hovmöller diagram in Figure 1 for more clarity. Does selecting a more offshore track affect CMES and MIOST results (Line 189)?

The Hovmöller diagram is computed along the coastline, therefore a track on Figure 1 would simply correspond to the coastline inside the parallelogram. To be clearer, I reformulated the caption of the Hovmöller diagram in this way: "Each column represents the sea level anomaly time series at a coastal location within the parallelogram shown in Figure 1, progressing from the southwesternmost point northward along the coast."

The strongest signal of a coastal trapped wave is typically found near the coast and weakens with increasing distance offshore. However, in order to be able to fully answer to the reviewer, I have also computed the Hovmöller diagram using a coastal track shifted 0.2° eastward in longitude (as explained below, this track is used to define the locations). A slightly weaker signal is observed. MIOST still shows better agreement with BLUELINK than CMEMS, although the phase speed differs more than in the coastal case. Shifting offshore is indeed suboptimal for altimetry, as the influence of off-shelf measurements increases in the optimal interpolation process.

[Figure]

- Lines 186–187 : that can be move to the figure caption.

Done

Phase Speed Computation :

- The computed phase speed appears to be underestimated since the dashed line does not cross SLA maxima for BLUELINK, and the maxima are difficult to identify in CMES data.I was wondering how does the Radon transform method compute phase speed in the absence of a clear propagation pattern?

Please see the answer to the next point

- Is the phase speed computed only for the first propagation, and does it remain consistent with subsequent propagations? For example, the CTWs propagating between 10/15/2023 and 11/01/2023 seem better captured by MIOST and CMES. Do these waves share the same phase speed?

There are few points to be clarified. First of all, the phase speed is not computed only for the first propagation: The Radon transform identifies all dominant linear features in a Hovmöller diagram, not just one. It reflects the combined effect of multiple propagating signals, rather than isolating a single phase speed.

Secondly, the Radon transform works by integrating values along lines at various angles (which correspond to different phase speeds). A stronger signal (such as the strongest CTW event identified by the reviewer) will contribute more intensity to the integral along the line that matches its slope. To highlight this, I have shifted the dotted line to match the strongest CTW event in the plot.

Thirdly, there is no absence of clear propagation pattern in our dataset, but rather higher or lower signal-to-noise ratio. This is demonstrated by the confidence intervals that can be computed as a result of the application of the Radon transform. This score is a form of normalized sharpness and gives an indication of how much the peak stands out from the noise in terms of standard deviations. A signal-to-noise ratio greater than 2 often suggests a significant signal, as it means the peak is more than 2 standard deviations above the mean, implying that the signal is quite distinct from the noise. In

this case, the BLUELINK and MIOST are above 2, respectively 2.87 and 2.29. The confidence score for CMEMS is 1.76, which confirms the results presented in the manuscript.

To support these statements, I show below the normalized sum of squares from the Radon transform (see Equation 4). A clear propagation pattern is visible in all three datasets, as indicated by a distinct dominant peak in each case. The differences in signal to noise ratio are reflected in how sharp the dominant peak is compared to nearby values, which represent other possible propagation angles, and by the presence of a smaller peak around 110 degrees in the two altimetry datasets.

[Figure]

.

- I suggest to compute SLA lagged correlations along the track from a reference point (e.g., Bermagui) . Then, estimate phase speed using a distance vs. lagged correlation plot.

I understand that this is an alternative method for estimating phase speed. However, based on the explanations I provided in the previous response regarding the Radon transform, I do not see the benefit of applying this alternative. As Almar et al. (2014) state: 'The accuracy [of the Radon transform in estimating wave speed] is fairly insensitive to wave characteristics, whereas the main limitations arise from the sampling scheme, specifically the number and density of wave gauges.' The use of lagged correlations to estimate phase speed would be appropriate if I were working with sparsely distributed tide gauge data. This is not the case here, as I am using a regularly spaced dense datasets.

*Almar, R., Michallet, H., Cienfuegos, R., Bonneton, P., Tissier, M. and Ruessink, G., 2014. On the use of the Radon Transform in studying nearshore wave dynamics. Coastal Engineering, 92, pp.24-30. https://doi.org/10.1016/j.coastaleng.2014.06.008*

EOF Analyses:

- Using EOFs to study CTWs might introduce biases, as the same propagation pattern could be projected onto different modes of variability. To better capture propagative patterns, consider using complex EOFs.

Thank you for this very useful suggestion. CEOF is now applied (on offshore masked maps as suggested later) and it helped indeed to improve, for example capturing more variance in the altimetry dataset. Methodology and results have been updated accordingly in the new manuscript.

- How does the other EOFs modes appear?

The second CEOF mode accounts for 9%, 13%, and 20% of the variance in BLUELINK, MIOST, and CMEMS, respectively. Below, I present its representation in terms of reconstructed signal, amplitude, and phase. It can be observed that this mode consists of oscillations similar to those of the primary CEOF, particularly in terms of period, but with lower amplitudes and a less distinct, noisier phase pattern. I suspect that a significant portion of the detected signal may be attributable to noise; however, lacking a solid scientific basis to support this claim, I have chosen not to include this discussion in the main manuscript.

[Figure]

- Consider presenting EOFs results in a "Hovmöller aesthetic," with distance on the x-axis, days on the y-axis, and EOF amplitude in color. This could help reduce figure size and enhance clarity.

In the new version, CEOFs results are presented in a "Hovmöller aesthetic" in the new Figure 6

- Merging Figures 6 and 7 may also be beneficial.

The figures related to the EOF analyses have completely changed to match the suggestions reported by the reviewer.

CTW characterizations:

I think the paper lacks a clear description of the CTWs observed in terms of period and spatial patterns. You could apply a complex EOF analysis on offshore-masked maps to estimate wave wavelengths an phase. Or Adding snapshots would illustrate how different products reproduce spatial patterns associated with CTWs.

Thank you for these suggestions. I applied all of them. Thanks to the application of the CEOF I have reported the spatial and temporal phase along the coast (Figure 7, lower panels) and the estimation of the dominant wavelength and period of the main CEOF. The snapshots of both the original filtered signal and the reconstructed signal from the first CEOF coinciding with the strongest CTW event are now added in Figure 7 and C1. The discussion has been updated accordingly.

---

## Author Comment (AC2)

Review of

Can satellite altimetry observe coastally trapped waves on sub-monthly timescales?

Marcello Passaro

General comments

The manuscript focuses on the potential of improved altimetry products to detect coastally trapped waves (CTW). So far, those waves were identified using tide gauge data that is very accurate with a high temporal resolution but sparsely spaced along the worlds coastlines. The manuscript successfully demonstrates that new products are able to show how CTWs propagate anticlockwise along the eastern Australian coast.

The author uses a combination of standard (correlations, EOFs) and (to me) less known techniques (image processing) to arrive at their conclusion. I find the argumentation convincing but I think the presentation of the results could be improved (see below).

I would like to thank the reviewer for the constructive feedback. Below, I provide a point-by-point response to the comments. The revised sections in the manuscript are highlighted in red.

Specific comments

Line 68: Remove parentheses for Woodham et al.

Done

Figure 1: I found the green rectangle a bit hard to see. Consider a different color and thicker lines.

A thicker line is used in the new version

Line 68: Can you say something about the temporal resolution of this dataset, and how this compares to the other datasets used in this study?

The temporal resolution of BRAN2020 is now reported: "The latest version, BRAN2020, is used. It simulates the period from 1993 to 2023 using a near-global, eddy-resolving ocean model with a 10-km spatial resolution and daily temporal resolution". Concerning the other datasets (altimetry), I have treated the issue in different sections, for example in the introduction: "Secondly, although these grids are currently released at a daily rate, previous literature has defined their effective temporal resolution to be about one month"

Line 71: what kind of sea surface height data is assimilated? Remote- sensing data I assume?

I now specify: "...and sea surface height (from satellite altimetry)"

Line 86: To be clear: The major difference between CMEMS and MIOST is the inclusion of the swath-altimeter data in MIOST? Both, MIOST and CMEMS use the MIOST technique to solve the mapping problem? I found that a bit confusing and would recomment choosing a different short name for what is now the MIOST product.

Your understanding is correct. I adopted the suggestion by renaming the MIOST product as "MIOST$_{SWOT+nadir}$", which is the same convention used in the following, related paper:

Ballarotta, M., Ubelmann, C., Bellemin-Laponnaz, V., Le Guillou, F., Meda, G., Anadon, C., Laloue, A., Delepoulle, A., Faugère, Y., Pujol, M.-I., Fablet, R., and Dibarboure, G.: Integrating wide-swath altimetry data into Level-4 multi-mission maps, Ocean Sci., 21, 63–80, https://doi.org/10.5194/os-21-63-2025, 2025.

Line 95: Here you assume that CTWs have periods between 7 and 29 days? Can you justify this? Have most of the CTWs observed so far fallen in that range? (Aydın and Beşiktepe, 2022, state that CTWs typicall fall into the 8-16d range and I'm wondering why you extend your range to 29 days.)

Aydın and Beşiktepe, 2022, is a paper analysing CTWs in the Black Sea. I use the same filtering as in Woodhman et al., 2013, which is focused on our same area of study and uses the same reanalysis: "In order to isolate the principal CTW frequencies, the Bluelink data were filtered using a fifth-order Butterworth bandpass filter, with frequency cutoffs (3 dB) at 0.035 and 0.15 cycles per day (cpd). This passes oscillations in the range 28.6–6.7 days."

Equation 7: T is the number of time steps but also denotes the transposed matrix X', right? Maybe choose another letter for the number of time steps (N?) to avoid confusion.

Done

Line 146: "The TG clearly shows...": is this from visual inspection only or is there more evidence?

I have rephrased in the following way: "A visual inspection of the TG data reveals oscillations with a period of approximately 10 days and amplitudes ranging from 5 to 10 cm."

Figure 2: Legends: A bit pedantic but could you show the TG first or last in the legend? As it is the ground truth. Also, one legend might be enough – the filtered/unfiltered can be in the titles for the two panels. The time step is days?

All the suggestions have been applied to the new version

Line 148: Have you already shown that the variability seen in the data is due to CTWs? Maybe based on previous literature? Or is it only an assumption at this point? Could the oscillations be caused by something else?

I added the following clarification: "For the purposes of this section, it is assumed that the observed oscillations represent the signature of CTWs, which are known to dominate sub-monthly variability in the study region (see Introduction). This assumption will be further examined in the following sections."

Line 172-173: In the previous paragraph you identified Bermagui as the TG with the lowest correlation of the filtered time series with the altimetry data. It seems a bit unfair using that one for the 2D validation. I understand that you choose the southernmost location and that it shouldn't matter as the shelf gets wider northward. But could you still justify that choice?

The justification is added in the form of the following paragraph: "Bermagui is chosen because it is the southernmost location in the domain, in order to highlight, through lag-correlation, the spatial footprint of coastally trapped waves traveling northward. However, the same statistics have been produced for every TG station, and the corresponding figures can be found in the Appendix"

Figure 5: For better readability consider giving the distance from first point in kilometers on the x-axis. Also, you could have a marker on the x-axis to show the approximate location of the tide gauges such that the Hovmöller diagrams are easier to relate to the maps in Figure 4. In addition, for consistency, use the same labels for the time (y-) axis as in the other figures, i.e. time steps (which are days, I presume?) instead of actual dates.

I have adopted both suggestions concerning x-axis and y-axis. I have not added the tide gauge locations, since I find this confusing given that the tide gauge data are not used in the Hovmöller diagram.

Line 193: The EOF analysis is very informative but I think it needs to be explained better, particularly the spatial EOFs. You can compare them to the correlation maps in Figure 4 at lag 0 –there, you clearly see a see-saw on the shelf which is consistent with the first spatial EOF (of Bluelink, at least).

I have strongly modified the EOF analysis in order to improve the explanations. I have now adopted the complex EOF (CEOF) analysis and updated both methodology and discussions. In this way, the results of the CEOF are now shown by considering phase, wavelength and period of the signals. Moreover, the recostructed signal is shown in an "Hovmöller aesthetic", to match this suggestion with the requests from the other reviewer. The CEOF analysis is also extended to the whole shelf, in order to better represent the spatial structures.

Also, the figures need to be improved:

Figure 6:

- Shouldn't either the EOF amplitude or the PC amplitude have a unit (m)? That would be useful and would make the results more physical.

The first CEOF is now shown in terms of reconstructed signal to make the results more physical, the amplitude has therefore "m" as unit.

- upper panel: what is the location index and how does it relate to the distance in Figure 5?

The distance is now shown in the same way for all the plots and in "km" as suggested by the reviewer

- lower panel: time steps is days?

Done

Figure 7: The information to be conveyed here is interesting but I wonder if you could do it differently? Instead of showing one panel for each day with all three datasets how about showing only three panels, one for each dataset. In the panels the EOFs could be shown for each time step (using different shades of blue/red/green) so that the anticlockwise rotation becomes very clear.

- as above, what is the location index and how does it relate to the distance in Figure 5?

- I think a unit is missing here

Figure 7 has now changed significantly and I tried to combine the suggestions coming from both reviewers. I now show the reconstructed signal over the entire shelf region from the three datasets from three specific days corresponding to the major CTW event. In the appendix, the same is shown also for the original filtered signal before the CEOF.

---

## Referee Report (RR1)

**Can satellite altimetry observe coastally trapped waves on sub-monthly timescales?**

The revised version of the manuscript shows clear improvements compared to the original submission. The clarifications introduced by the authors significantly enhance the readability of the text and the methodological explanations are now much easier to follow. A notable strength of the revision is the integration and discussion of the Complex Empirical Orthogonal Function analysis. This addition provides a more robust assessment of the spatio-temporal characteristics of the coastally trapped waves and demonstrate the propagating nature of the detected signal. Overall, the manuscript is well-constructed, and I recommend publication after the addressing of the following concerns.

**Introduction:**

Please better explain why accurately detecting CTWs is crucial for describing and predicting coastal circulation, upwelling processes, and associated ecosystem impacts. This aspect is currently mentioned in the conclusion but should also be highlighted earlier to motivate the study.

**Data:**

- You add that the model is eddy-resolving, but a resolution of 10 km makes the model eddy-permitting in the study region.

**SWOT Coverage and Sampling:**

- Please indicate where the SWOT swaths intersect the continental shelf in your study region. Adding this information directly in Figure 1 would greatly help interpretation.
- In Figure 2 (Coffs Harbour), does the time series correspond to a location directly sampled by SWOT? If so, it would be useful to compare results with a location not sampled by SWOT to assess sensitivity to swath coverage.

**Spectral Content**

- The CMEMS and MIOST time series appear to show virtually no energy at periods shorter than 20 days. How do you reconcile this with the periods of 14 days (MIOSTSWOT+nadirs) and 18 days (CMEMS) obtained from the CEOF computation?
- -Both the spectra and Figure 3 show a peak around 22.5 days. This feature is not currently discussed. Please clarify whether this peak corresponds to CTWs or to another process.

- -In Figure 4, the correlation with Bermagui does not appear to be significant for any of the products, even though Figure 8 shows correlations exceeding 0.8. Could you provide an explanation for this discrepancy?
- -Line 213: you state that the first CEOF mode explains 71% of the total variance of the filtered signal. Please clarify whether this refers only to the filtered signal or to the total variance.

**CEOF** modes:**

I agree with your answer and is not necessary to add the analysis of modes higher than one in the manuscript.

**Conclusions**

I am globally satisfied by the answers given by the authors to my first review and I recommend the publication after minor corrections.

---

## Author Response (AR2)

**REVIEWER 1**

Can satellite altimetry observe coastally trapped waves on sub-monthly timescales?

Marcello Passaro

Revised version

**General comments**

The author has addressed all my comments satisfactorily. In response to the first reviewers comments, the EOF analysis has been replaced with a complex EOF analysis which is better suited for propagating patterns. I think this is very beneficial for the analysis and characterisation of the observed CTWs.

I have no major comments and only a few minor comments that, I think, might help to improve the clarity of some of the figures.

**Specific comments**

Section 3.3 Suggest to rename this section to "Radon transform and phase speed computation"

**Suggestion accepted**

Equation 6: Xc is not used in the subsequent equations, so why define it here?

The definition of Xc is correct in Equation 6, but should have been used consistently in Equations 7 and 9. This is now corrected, thanks for spotting the issue!

Figure 3c: move legend to the left for better visibility?

**Suggestion accepted**

Line 194: anti-correlated but not statistically significant, or only partly!

I rephrased as "Shelf locations north of 31 degrees S are anticorrelated with Bermagui at lag 0, although some of these correlations are not statistically significant,..."

The following suggestions might also be just personal preference but I think they could contribute to more clarity in the figures.

Figure 4: you actually only need one colorbar (or one per row) as they all have the same range.

I prefer to keep the colorbars in each subplot. In my experience, subplots are often copypasted or cropped in presentations or other contexts without their corresponding colorbars, which can lead to misinterpretation.

Figure 5 and 6 (upper row): I would find it more intuitive to have the distance as the y-axis (kind of mirroring the coast) and time as y-axis. Also, as in Figure 5, one colorbar might suffice.

I prefer to keep time on the y-axis to facilitate easier comparison with key references in the literature, particularly Woodham et al. (2013). Their Hovmöller plots, which I include below as a screenshot, also adopt this convention.

Figure 6, lower row: Would it be more intuitive to show the phase in degrees? And, particularly for the right panel, how about keeping the phase between 0-360 (thinking degrees here) to even more emphasize the oscillations?

While this wrapped representation can visually emphasize oscillatory behavior, this is already clear from the Hovmöller diagram. My analysis in the text related to the lower row relies on the unwrapped phase to compute key physical quantities such as wavelength and period. Unwrapping the phase avoids artificial discontinuities and allows for accurate spatial and temporal derivatives, which are essential for quantifying wave propagation characteristics.

Figure 7: same as for Figure 4.

**See previous comments**

**REVIEWER 2**

The revised version of the manuscript shows clear improvements compared to the original submission. The clarifications introduced by the authors significantly enhance the readability of the text and the methodological explanations are now much easier to follow. A notable strength of the revision is the integration and discussion of the Complex Empirical Orthogonal Function analysis. This addition provides a more robust assessment of the spatio-temporal characteristics of the coastally trapped waves and demonstrate the propagating nature of the detected signal. Overall, the manuscript is well-constructed, and I recommend publication after the addressing of the following concerns.

**Introduction:**

Please better explain why accurately detecting CTWs is crucial for describing and predicting coastal circulation, upwelling processes, and associated ecosystem impacts. This aspect is currently mentioned in the conclusion but should also be highlighted earlier to motivate the study.

The following lines are added to the introduction: CTWs can interact strongly with local circulation patterns, inducing alongshore currents and significant water mass displacement (Bailey et al., 2022). By causing vertical displacements of the pycnocline, CTWs affect coastal upwelling patterns and thereby influence nearshore productivity (e.g., Echevin et al., 2014). Monitoring and understanding this phenomenon can therefore be useful for improving predictions of biological productivity in coastal systems (Körner et al., 2024).

**Data:**

- You add that the model is eddy-resolving, but a resolution of 10 km makes the model eddy-permitting in the study region.

The statement is taken from the related references and it is likely related to the "near-global" coverage. However, to avoid misunderstandings, I have removed "eddy-resolving".

SWOT Coverage and Sampling:

- Please indicate where the SWOT swaths intersect the continental shelf in your study region. Adding this information directly in Figure 1 would greatly help interpretation.

We are using data from the 21-day science orbit phase of the SWOT mission. During this period, SWOT provides near-global coverage of the ocean between approximately 78°N and 78°S. As a result, nearly all of our study locations are intersected by SWOT swaths. While the spatial coverage is extensive, the primary limitation of SWOT during this phase is temporal: each location is revisited only once every 21 days. Therefore, the constraint is not spatial coverage, but rather the temporal resolution of the observations.

- In Figure 2 (Coffs Harbour), does the time series correspond to a location directly sampled by SWOT? If so, it would be useful to compare results with a location not sampled by SWOT to assess sensitivity to swath coverage.

**Please see the answer to the previous comment**

**Spectral Content**

- The CMEMS and MIOST time series appear to show virtually no energy at periods shorter than 20 days. How do you reconcile this with the periods of 14 days (MIOSTSWOT+nadirs) and 18 days (CMEMS) obtained from the CEOF computation?

I respectfully disagree with the statement that there is "virtually no energy" at periods shorter than 20 days. To clarify this point, I have produced a special version of Figure 3D for your benefit, in which I have added vertical lines at 14 and 18 days.

As shown in this version, both datasets do exhibit spectral energy at these periods. Therefore, I do not see a contradiction that requires reconciliation.

-Both the spectra and Figure 3 show a peak around 22.5 days. This feature is not currently discussed. Please clarify whether this peak corresponds to CTWs or to another process.

The apparent peak around 22.5 days is not a physical feature, but rather an artifact introduced by the filtering process. Specifically, when generating the PSD of the filtered signals, the last discretized period value before the cutoff is 22.5 days. This can give the impression of a peak at that location.

To clarify, this feature does not correspond to coastally trapped waves (CTWs) or any other physical process. This can be appreciated in Figure 3C, which shows the PSD of the unfiltered signals: no peak is visible around 22.5 days in that panel. The presence of the feature in the filtered PSD is purely a consequence of the bandpass filter design and the discretization of the frequency axis.

-In Figure 4, the correlation with Bermagui does not appear to be significant for any of the products, even though Figure 8 shows correlations exceeding 0.8. Could you provide an explanation for this discrepancy?

There is no Figure 8 in the manuscript, but I assume the reviewer is referring to Figures 3A and 3B, where the correlations are presented. Figure 4 shows the correlation of the filtered signals, which corresponds to the analysis in Figure 3B. As shown there, the only dataset with a strong correlation at the closest shelf point is BLUELINK, which is consistent with what is seen in Figure 4. Neither CMEMS nor MIOST\$\_{SWOT+nadirs}\$ shows correlations exceeding 0.8, not even in the full signal analysis shown in Figure 3A. Therefore, I do not see a discrepancy between the figures.

-Line 213: you state that the first CEOF mode explains 71% of the total variance of the filtered signal. Please clarify whether this refers only to the filtered signal or to the total variance.

**I changed "total variance" to "variance of the filtered signal"**

CEOF modes: I agree with your answer and is not necessary to add the analysis of modes higher than one in the manuscript.

Conclusions I am globally satisfied by the answers given by the authors to my first review and I recommend the publication after minor corrections.